# Compressive and Flexural Strengths of Mortars Containing ABS and WEEE Based Plastic Aggregates

**DOI:** 10.3390/polym14183914

**Published:** 2022-09-19

**Authors:** Youssef El Bitouri, Didier Perrin

**Affiliations:** 1LMGC, IMT Mines Ales, Univ Montpellier, CNRS, F-30100 Ales, France; 2PCH, IMT Mines Ales, F-30100 Ales, France

**Keywords:** plastic-aggregate, ABS, mortar, mechanical properties, flexural strength, rupture

## Abstract

The incorporation of plastic aggregates as a partial replacement of natural aggregates in cementitious materials is interesting in several ways. From a mechanical point of view, the partial substitution of sand with plastic aggregates could improve some properties (e.g., ductility, thermal insulation). This paper deals with the mechanical strength of mortars containing plastic aggregates as a partial replacement of sand. Part of the volume of sand in cement mortars is substituted with plastic aggregates which originate from WEEE (Waste from Electrical and Electronic Equipment) and consist of a mix of ABS (acrylonitrile-butadiene styrene), HIPS (high impact polystyrene) and PP (Polypropylene), or of monomaterial ABS from WEEE sorting. Three rates of replacement (by volume of sand) were tested: 10%, 15% and 30%. Mechanical tests were performed according to European standard EN196-1. The results show that compressive and flexural strength decrease with rate of replacement, but remain satisfactory for structural purposes. In addition, the density of mortar is reduced with the incorporation of plastic aggregates. The decrease of mechanical strength is mainly due to the weak bond between cement paste and plastic aggregates leading to the increase of porosity. Furthermore, it appears that mortars containing plastic aggregates could present a ductile rupture.

## 1. Introduction

The recycling and valorization of plastic wastes is a growing challenge from both an environmental and economic point of view. With more than 300 million tons produced each year around the world [1], recycling and valorization solutions must be proposed. Among the plastic wastes that could pose problems, we can cite the Wastes from Electrical and Electronic Equipment (WEEE). This type of polymer wastes contains a complex mixture of materials, some of which are hazardous [2]. Different types of thermoplastic polymers such as styrenic matrices (e.g., ABS for Acrylonitrile Butadiene Styrene, HIPS for High Impact Polystyrene, PP for Polypropylene) can be found in these wastes.

One of the interesting options for reusing these wastes is their incorporation into construction materials such as cementitious materials (mortars, concretes). This solution could present several possibilities of interests. In addition to the possibility of recovering large volumes and decreasing the landfilling, it would make it possible to confine the waste and limit its dissemination in the environment. The partial substitution of mineral aggregates by plastic aggregates would also make it possible to reduce the pressure on natural resources. Furthermore, the cementitious materials formulated with plastic aggregates could have interesting properties. In fact, thermal insulation performance can be improved by the addition of plastic aggregates, such as polyethylene (PET) [3], as well as the mechanical properties, especially ductility [4,5,6].

The incorporation of aggregates from plastic wastes is well documented in the scientific literature, and some critical reviews are available [7,8]. The data are quite dispersed in terms of the substituted component (fine or coarse aggregates), the type of plastic, the substitution rates and the pretreatments carried out on the plastic before incorporation into the cementitious materials [9,10,11,12]. Plastic wastes frequently studied are polyethylenes terephthalate (PET), high-density polyethylenes (HDPE), polyvinyl chlorides (PVC) and polypropylenes (PP). The pretreatments are sometimes used to improve the bond with the cement paste. Gamma radiation treatment of plastic polymers is the most widely used [10]. Other types of treatment based on atmospheric plasma were performed [9,11]. These treatments aim to physically modify the surface roughness while adjusting the nature of the functional groups of plastic polymers according to the surface energy of the cement pastes.

In general, the available data point out the deleterious effect of partial substitution of sand by plastic aggregates on the mechanical properties of concretes [13,14,15,16]. Here, we report only some studies performed on mechanical properties of mortars. Abed et al. [17] tested five different waste PET weight fractions of 0, 5, 15, 25 and 50% as a replacement of sand in cement mortar with constant cement content (525 kg/m^3^) and water-to-cement ratio of 0.48. In general, it was found that compressive and flexural strengths were reduced as waste PET incorporation increased. The mortar containing 25% waste PET appears to be a lightweight mortar with satisfactory mechanical properties suitable for structural purposes. Ghernouti and Rabehi [18] reported a reduction of compressive and flexural strengths according to the increase in replacement rate of sand by plastic aggregates. These mechanical strengths remain almost close to those of the reference mortar for replacement rates below 20% by volume. According to Carneiro et al. [19], the substitution of sand by shredded PET in polymer mortar containing epoxy resin (12% by weight) as binder contributes to a decrease in both flexural and compressive strength. For flexural strength, the decrease is of about 48% for PET content of 20% (by weight), while compressive strength decreases by 59%. The authors explain this overall decrease by the poor bond between shredded PET waste and the epoxy matrix. Merlo et al. [20,21] studied the reuse of PVC deriving from WEEE as a partial substitute for sand to produce lightened mortars. The volume replacement rates of sand by plastic aggregates of 5, 10, 15 and 20% were used. The results obtained with ordinary Portland cement showed a strong decrease in both compressive and flexural strengths. The authors attribute this decrease to the mechanical characteristics of PVC and to the poor adherence between cement paste and plastic aggregates. Recently, the same authors published another study carried out with replacement rates of up to 90% (in volume) [21]. The decrease of the mechanical properties (compressive and flexural strengths) is of about 40% with a replacement rate of 15% vol. It should be noted that this decrease could be due not only to the poor adherence between cement paste and plastic aggregates, but also to the granular characteristics of the latter. In fact, the fines content of plastic aggregates was not sufficient to achieve the optimum packing leading to the minimum of porosity. The fineness modulus of plastic waste used was 2.5, while for standard sand, it was 3.0 [21].

Despite the large number of studies on the reuse of plastic wastes as aggregates in cementitious materials, only a few studies have focused on the styrenic materials encountered in WEEE, such as acrylonitrile-butadiene styrene (ABS) and high-impact polystyrene (HIPS) [4,22]. It can be noted that Makri et al. [22] investigated the physical and mechanical properties of cement mortars, partially replaced with ABS-based aggregates from plastic housing of LCD screens. The replacement percentages used were 2.5%, 5%, 7.5%, 10% and 12.5% (the authors do not specify whether the replacement rate is by mass or by volume), while the water-to-cement (w/c) ratio was maintained constant at 0.5. The obtained results show a decrease in compressive strength except for the replacement rate of 7.5% and 10%, which exhibited an increase by 15.4% and 7.8%, respectively. The elastic modulus (at 28 days) decreases by 39%, 48%, respectively, for the replacement rates of 2.5 and 5%, while only 19%, 24% of decrease is recorded for the replacement rate of 7.5% and 10%, respectively [19]. Another more recent article regarding physical properties and microstructure of WEEE-based plastic aggregate mortars made with acrylonitrile-butadiene-styrene (ABS), polycarbonate (PC), polyoxymethylene (POM), polyethylene (PET) and ABS/PC blend waste was produced by Kaur and Pavia [23]. The loss of compressive strength is significant with PET and POM-based aggregates, up to 42% with 20% (by volume) of replacement. At 5% of replacement (by volume), the PC, ABS and ABS/PC plastics increase the compressive strength by 6–15%. At 20% of replacement, the loss of compressive strength is of about 25%. The reduction in flexural strength is less pronounced than the compressive strength reduction. At 20% of sand replacement, the reduction of flexural strength is of about 21% for PET and 36% for PC and ABS/PC. Furthermore, the authors consider that the mechanical strengths (compressive and flexural strengths) met the standard strength requirements for masonry, rendering and plastering mortars. Moreover, the plastic aggregates increase the ability of the mortar to absorb deformation upon the application of stresses, retarding failure and transforming the brittle failure typical of cement mortars into a ductile failure. The authors notice that the particle size, shape and surface characteristics of the plastic particles have a significant impact on the properties of the resulting material. Finally, the results highlight the optimal performance from ABS scraps smaller than 2 mm, which hardly increases the hygric properties of mortars t, and shows high resistances and evidence of unbroken binding at the interfaces [23]. Therefore, by improving the compressive and flexural strengths, the use of recovered plastics from WEEE, such as styrenic thermoplastic materials mainly based on ABS, as recycled aggregates in partial substitution of sand in cement mortar could potentially prove a useful recycling alternative for plastic waste.

Therefore, it appears that the partial substitution of sand with plastic aggregates in cement mortars leads to a decrease in strength, especially with high replacement rates [14,18,20,21,23,24,25]. The loss of strength is often attributed to the weak cement paste/plastic aggregates bond induced by the hydrophobic nature of the plastic. However, the results in the literature are dispersed and depend not only on the nature of plastic aggregates, but on the particle size distribution of these aggregates. For instance, the maximum size of plastic particles is often different and sometimes exceeds that used in mortar sands (maximum 4 mm) [21], as well as the fineness modulus [14,18,20,21,25]. For the results to be comparable, it would be more appropriate to formulate mortars with plastic aggregates, whose the particle size distribution is close to that of sand according to the requirements of a standard such as the European standard EN 13139 or EN 196-1 [14,26,27]. In addition, only a few studies concern the styrenic materials encountered in WEEE (ABS).

In this work, the effect of a partial substitution of sand by plastic aggregates from styrenic materials originating from WEEE (Waste from Electrical and Electronic Equipment) on compressive and flexural strength of cement mortars is investigated. The plastic aggregates were grinded and sieved in order to obtain a range size of particles between 0 and 2 mm close to that of standard sand [26]. Furthermore, the quality of adherence between cement paste and these aggregates was investigated qualitatively through scanning electron microscopy (SEM) observations coupled with Energy-Dispersive X-Ray Analysis (EDX).

## 2. Materials and Methods

This study was focused on styrenic materials encountered in WEEE, which are principally represented by ABS (acrylonitrile-butadiene styrene), HIPS (high impact polystyrene) and PP (Polypropylene). Samples were provided by Suez Group (Berville-su-Seine, France) and consist of plastics from WEEE (mix of ABS, HIPS and PP, Figure 1a) and mono-material ABS from WEEE sorting (Figure 1b). These samples were cut with a Retsch^®^ SM300 grinder (Haan, Germany) at 5000 rpm using a 2 mm grid. The obtained material is a mix of particles with different sizes, as shown in Figure 1. The waste absolute density was evaluated through a pycnometer (Micromeritics AccuPyc 1330, Micromeritics Instrument Corporation, Norcross, GA, USA). Density measurements were performed in triplicate (Table 1).

The CEN standard sand was used to prepare the mortars. It is a natural siliceous sand consisting of rounded particles and has a silica content of at least 98%. According to the European standard (EN-196-1), CEN standard sand has a specific particle size distribution ranging between 0.08 and 2.00 mm. The particle size distribution of CEN standard sand and plastic wastes was determined through vibratory sieving using sieves of 2 mm, 1.6 mm, 1 mm, 0.5 mm, 0.16 mm and 0.08 mm, and plastic particles larger than 2 mm were removed. These measurements were performed in triplicate. Particle size distributions of sand and plastic aggregates are shown in Figure 2.

As shown in Figure 2, WEEE and ABS-based aggregates exhibit the same granulometry after grinding. The fines content of these aggregates is lower than standard sand.

An ordinary Portland cement (CEM I/42.5 N) was used to prepare the mortar mixtures.

The mortar mixtures were prepared according to the European standard (EN 196-1). The composition of the different mortars is provided in Table 2. Plastic aggregates were used as a partial replacement of sand volume with three replacement rates: 10, 15 and 30%.

The procedure of mixing was performed according to the following sequence: the water and the cement are placed into the bowl and mixed at low speed during 30 s. The sand is added steadily during the next 30 s, and the mixing is continued at high speed for additional 30 s. The mixer is then stopped for 90 s. During the first 30 s, the mixer walls are scraped to homogenize the mortar, and the mixing is continued for 60 s at high speed.

For each mortar composition, three specimens were prepared and placed in prismatic molds 40 × 40 × 160 mm^3^. After 24 h, all samples were demolded and immersed in water at 20 °C for 28 days.

The chemical types of both WEEE and ABS plastic aggregates were checked by a Vertex 70 FT MIR spectrometer from Bruker (Billerica, MA, USA) with an ATR unit used (Figure 3). The used resolution was of 4 cm^−1^, 32 scans for background acquisition and 32 scans for the sample spectrum. Spectra were acquired from 4000 to 400 cm^−1^ and analyzed using the OPUS software provided with the spectrometer. Most of the samples were directly analyzed on the crystal. Standard samples and dirty waste samples were previously cleaned with ethanol.

The FTIR spectroscopic analysis with ATR mode provides the identification as the family of styrenics through HIPS and ABS polymers. The most remarkable patterns of styrenics are the two aliphatic CH stretching signals at 2920 cm^−1^ and 2850 cm^−1^ as well as the 3–5 cm^−1^ aromatic CH stretching signals between 3000 and 3100 cm^−1^ and the two aromatic CH waging at ≈700 cm^−1^ and ≈750 cm^−1^, the second one being thrice as small as the first one. The main difference between ABS and HIPS is the stretching of the carbon–nitrogen triple bond of acrylonitrile, which produces a fairly weak, but sharp signal at 2237 cm^−1^, a very specific location. Moreover, the other signal specificities between the ABS and the HIPS styrenic family was found at 695 ± 2 cm^−1^ for PS-based samples, but at 700 ± 3 cm^−1^ for SAN-based polymers. It can be deduced that the presence of acrylonitrile in the polymer chains affect the aromatic CH vibrations.

Thus, styrenic-based polymers from WEEE plastics ((a) in Figure 1) were identified as HIPS and ABS, while (b) was checked as an ABS polymer after ABS WEEE sorting.

Flexural and compressive strength evaluations were performed with according to the European standard (EN 196-1) [26]. Flexural strength was determined using the three-point loading method. For each mortar composition, flexural strength was measured on three specimens. Then, compressive strength is measured on halves of the prism broken on flexural testing using 3R RP400E-425kN (Montauban, France). These measurements were performed on six specimens.

In addition to mechanical tests, the polished sections of mortar were observed using Scanning Electron Microscopy (SEM Quanta 200 FEG (FEI Company, Hillsboro, OR, USA) with Energy-Dispersive X-Ray Analysis (EDX). Direct optical microscopy observations of the polished sections were performed with a Leica Laborlux 12 POL S optical microscope equipped with a 1600 × 1200 pixels mono-CDD Sony digital camera.

## 3. Results and Discussion

The aim of the mechanical characterization is to evaluate the effect of the partial substitution of sand by plastic aggregates (WEEE and ABS). The results are shown in Figure 4. The loss of compressive and flexural strength is represented in Figure 5.

These results show that the partial substitution of sand by plastic aggregates decreases both the compressive and the flexural strength. Furthermore, it appears that the compressive strength is more affected than the flexural strength, which is in accordance with literature [8,28,29,30]. For WEEE-based mortars, the compressive strength is decreased by about 11.7% for 10%v, 11.3% for 15%v, and 21% for 30%v. For ABS-based mortars, the compressive strength is reduced by 14.4% for 10%v, 22.8% for 15%v, and 23.5% for 30%v.

For ABS-based mortars, the compressive strength was slightly lower than that of WEEE-based mortars for replacement rates of 10%v and 30%v. Concerning the flexural strength, there is no significant difference between WEEE and ABS. It is interesting to note that the strength results of ABS-based samples were more dispersed than those of WEEE-based samples.

Concerning the flexural strength, the loss ranges between 8 and 17%. Mortars containing ABS-based aggregates show a flexural strength slightly better than that of WEEE-based aggregates. In addition, even if the displacement was not measured to estimate the fracture energy, it would seem that the ductility of ABS-based mortars is improved compared to that of other mortars, including the reference without plastic aggregates, as shown in Figure A1. In fact, the rupture is manifested by the propagation of a bending crack in the middle of the specimen without the specimen separating into two pieces, whereas for the other mortars, the rupture is sudden and without warning signs, which characterizes a brittle fracture under mode I. Further investigation with a complete mechanical characterization is required to examine this observation. In fact, if this increase of ductility is confirmed, the partial substitution of sand by ABS-based aggregates will be very interesting from a mechanical point of view (e.g., to limit the effect of drying shrinkage).

Furthermore, it has to be kept in mind that even if the partial substitution of sand by plastic aggregates leads to a decrease of compressive and flexural strength, the values of these strengths remain acceptable from a practical point of view. Moreover, a decrease in the self-weight of concrete could be obtained by the decrease of sand proportion. In fact, as shown in Figure 6, the decrease in density is of about 8% for WEEE and 11% for ABS at 30%v substitution.

The decrease in mechanical properties could be explained by different aspects. As shown in Figure 2, the particle size distributions certainly have slight differences, but they could lead to an increase in porosity in the hardened mortar. It should be noted that the initial compactness is the same for all the mixtures (of about 74.3%) since the substitution is performed by volume. In addition, the mechanical properties of plastic aggregates and sand are different. Finally, given the hydrophobic character of plastic aggregates, the quality of the cement paste/aggregate interface could be deteriorated, compared to that of sand, which can have epitaxial properties with the cement paste, i.e., the growth of cement hydration products on the aggregate surface and the chemical reaction between aggregate and cement. In order to evaluate the quality of this interface, SEM and optical observations were performed.

As shown in Figure A2, cement matrix/plastic or sand aggregate interfaces display a decohesion zone. This decohesion is more visible with plastic aggregate. Using EDX analysis in different zones (Figure 7 and Figure 8), the disturbed zones around plastic aggregates (WEEE and ABS) seem to be slightly more important than those around sand. The thickness of the disturbed zone is indicated by the slope of the signal drop of the marker element (C for plastic, Si for sand and Ca/Si for cement matrix) between the aggregate and the matrix (Figure 7 and Figure 8).

The difference between the particle size distribution of the reference mortar and the mortars containing plastic aggregates, and the quality of matrix/aggregate interface could lead to an increase in the porosity of mortar containing plastic aggregates (WEEE and ABS mortars). In fact, using the measured mechanical strengths and the extended Zheng model proposed by Chen et al. [31], the porosity of each mortar was assessed. As shown in (Figure 9), the porosity of the reference mortar is of about 5%, 10% for mortar with 10%v (ABS and WEEE), 13% and 9% for mortar with 15%v of ABS and WEEE, respectively, and 13% for mortars with 30%v of ABS and WEEE. It thus appears that porosity increases with increasing volume substitution. This increase of porosity is well correlated with the loss of strength.

The results obtained in this study are comparable to those found in the literature [14,17,20]. The compressive and flexural strength obtained with partial substitution of sand by plastic aggregates appear to be very interesting. In addition, the loss of mechanical strength observed on the mortar could be less important on the concrete.

## 4. Conclusions

The paper aimed to examine the effect of partial substitution of sand with plastic aggregates originated from WEEE on the mechanical strength of cement mortar. These plastic aggregates consist of a mix of ABS, HIPS and PP (WEEE-based aggregates) or only ABS from WEEE sorting (ABS-based aggregates). According to the obtained results, it appears that the partial substitution of sand leads to a decrease in strength.

For WEEE-based mortars, the compressive strength decreases by 11.7%, 11.3% and 20.8%, while the flexural strength decreases by 16.8%, 9.8% and 17%, respectively with 10%v, 15%v and 30%v of plastic aggregates. For ABS-based mortars, the loss of compressive strength is, respectively, of 14.4%, 22.8% and 23.6% with 10%v, 15%v and 30%v of plastic aggregates. The flexural strength of ABS-based mortars is reduced by 8.8% at 10%v, 13.7% at 15%v and 14.5% at 30%v. It can be noted that the flexural strength of cement mortar is less affected by the plastic aggregate replacement than the compressive strength, especially for ABS-based mortars.

The loss of strength can be explained by the increase of porosity induced mainly by the quality of plastic aggregate/matrix interface which is more porous than sand/matrix interface. Moreover, the density of the cement mortar is reduced by the substitution of sand by plastic aggregates.

Despite the loss of strength which remains acceptable from a practical point of view, the cement mortars containing plastic aggregates show a facies of rupture different from the reference mortar, suggesting an increase of ductility. Further investigations should be performed to examine this increase of ductility. In particular, a physical pre-treatment based on cold plasma that would attenuate the hydrophobic character of plastics could potentially lead to a better cohesion of the mixture, thus improving the properties of the final sample.

## Figures and Tables

**Figure 1 polymers-14-03914-f001:**
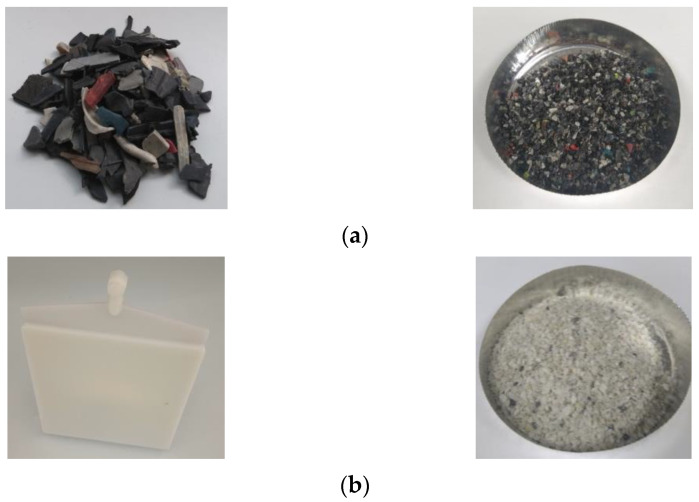
WEEE plastics (**a**) and ABS from WEEE sorting (**b**)-based wastes before and after grinding.

**Figure 2 polymers-14-03914-f002:**
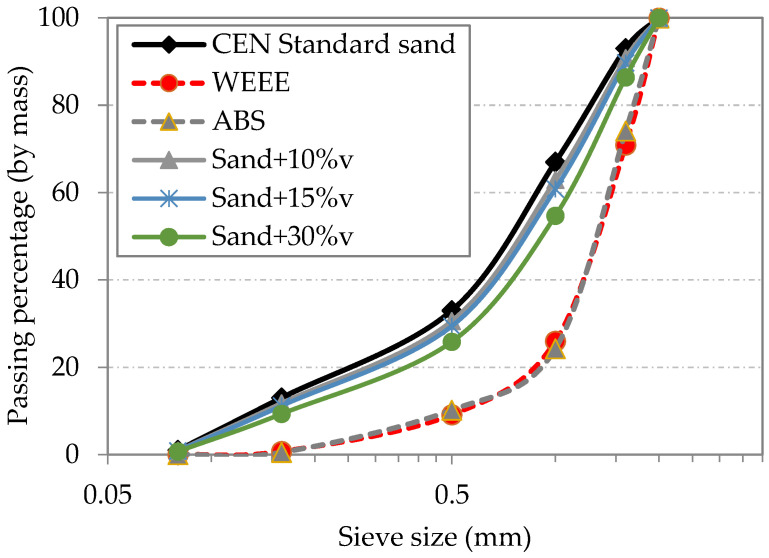
Particle size distribution of standard sand and plastic aggregates.

**Figure 3 polymers-14-03914-f003:**
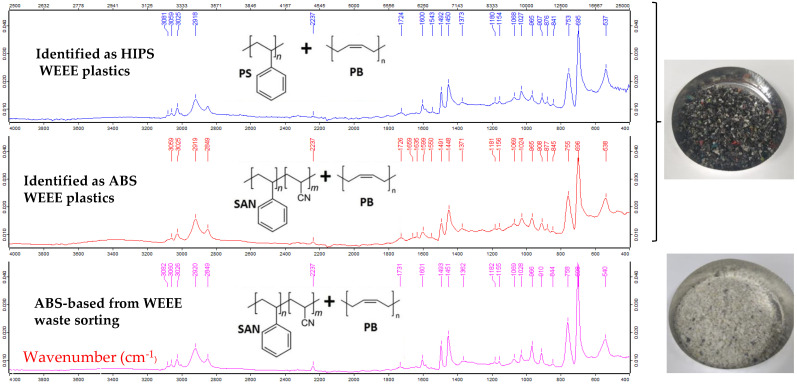
Identified FTIR with ATR mode as ABS-HIPS-based WEEE wastes (denoted (a) in Figure 1), and as ABS-based from WEEE waste sorting (denoted (b) in Figure 1).

**Figure 4 polymers-14-03914-f004:**
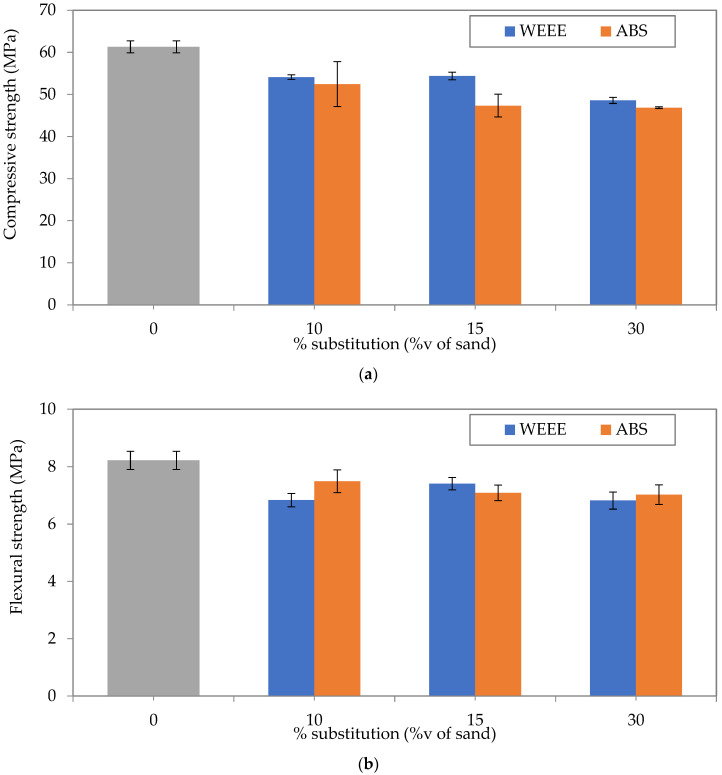
Compressive (**a**) and flexural (**b**) strength as a function of volume substitution of sand by plastic aggregates.

**Figure 5 polymers-14-03914-f005:**
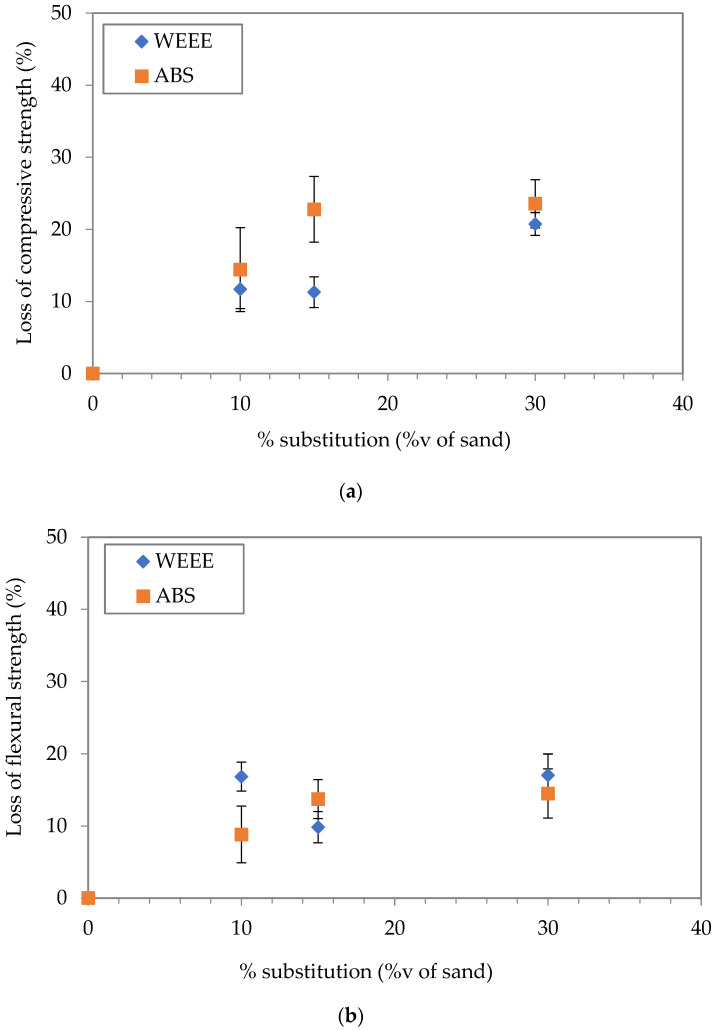
Loss of compressive (**a**) and flexural (**b**) strength compared to reference mortar without plastic aggregates.

**Figure 6 polymers-14-03914-f006:**
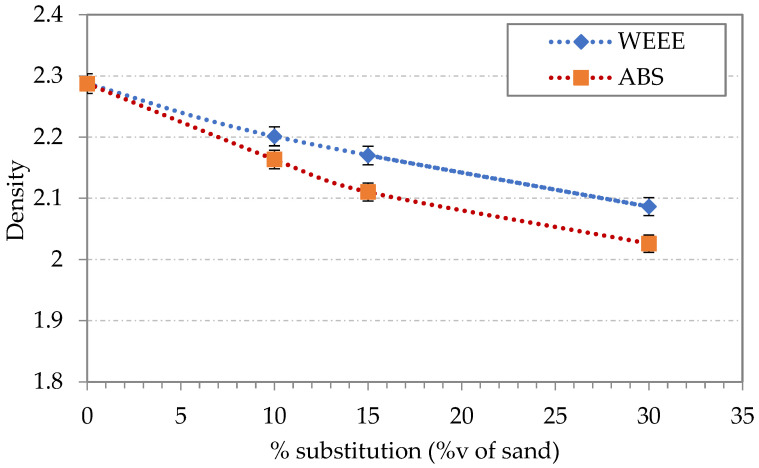
Density of mortars specimens as a function of volume substitution of sand by plastic aggregates.

**Figure 7 polymers-14-03914-f007:**
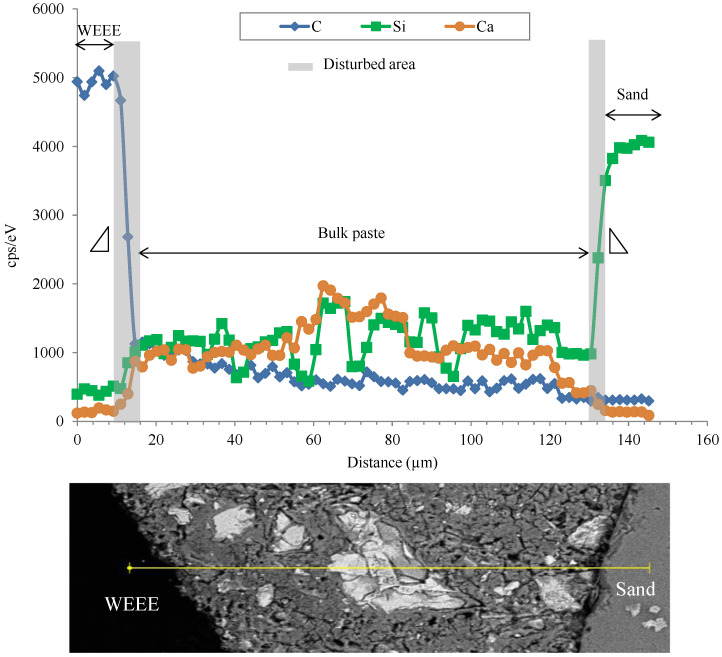
An example of Energy-Dispersive X-Ray analysis (EDX) of WEEE/cement matrix and sand/cement matrix interface.

**Figure 8 polymers-14-03914-f008:**
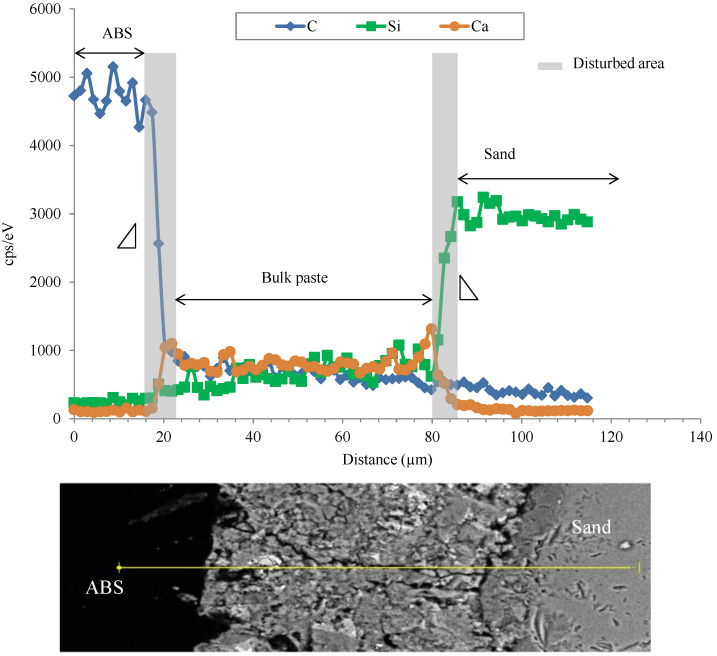
An example of Energy-Dispersive X-Ray analysis (EDX) of ABS/cement matrix and sand/cement matrix interface.

**Figure 9 polymers-14-03914-f009:**
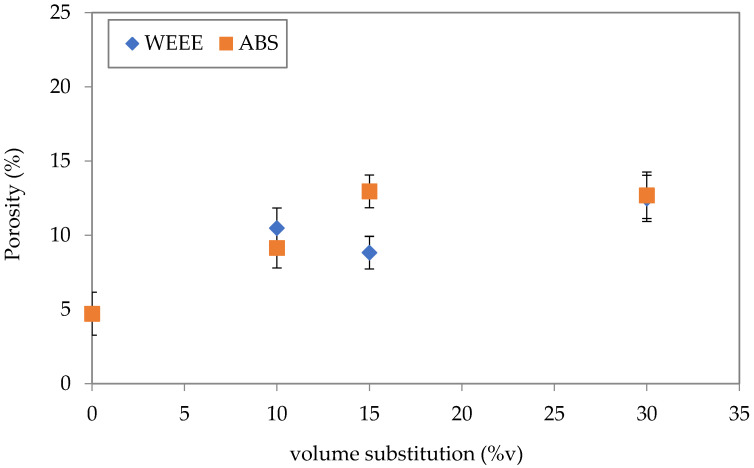
Porosity of mortar as a function of volume substitution.

**Table 1 polymers-14-03914-t001:** Absolute density of materials.

	Cement	Sand	WEEE	ABS
Measured density (g/cm^3^)	3.22	2.65	1.07	1.05

**Table 2 polymers-14-03914-t002:** Composition of the different mortars.

Sample	Cement (g)	Water (g)	Sand (g)	Plastic (g)
Reference	450	225	1350	0
WEEE10	450	225	1215	54.5
WEEE15	450	225	1147.5	81.8
WEEE30	450	225	945	163.5
ABS10	450	225	1215	53.6
ABS15	450	225	1147.5	80.4
ABS30	450	225	945	160.8

## Data Availability

Data archived on a hard drive by the authors.

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
