# Peer review of "Compressive and Flexural Strengths of Mortars Containing ABS and WEEE Based Plastic Aggregates"

_polymers, 2022, doi:10.3390/polym14183914_

Round 1

Reviewer 1 Report

Please find attached the file with comments.

Author Response

Dear reviewer, thank you for the quality of your comments and the time spent working on our article. You will find a complete response to your comments, hoping that they satisfy you.   Best regards,   Prof. Didier PERRIN

Reviewer 2 Report

Recycling of some solid wastes in civil engineering materials is a promising way to solve the environmental problems and resource waste. Therefore, the research topic is important and interesting. The authors are suggested to further revise some issues. Some suggestions:

1. The influence of different types of waste on the mechanical properties of cement mortar has not been clearly introduced in the section of Introduction. The statement should be straightforward, for example, “The mortar containing 25% waste PET appears to be most interesting mixture.”, What does it mean? And different types of waste have different effects on the mechanical properties of cement mortar, the authors should try to find some explanations.

2. The aim of this research is also to evaluate the effect of thermoplastics waste based aggregates on the mechanical properties of cement mortar. The differences between this study and previous studies should be clarified in detail in the section of Introduction.

3. Provide more information about the raw materials, for example, WEEE is a mixture, what are the proportions of the different components?

4. There is a citation error in line 125.

5. According to the Figure 2, the gradation of Sand+30%v is different from others. Does it have no effect on the test results of mechanical properties?

6. Please add unit to Table 1.

7. Try to improve the presentation of Figure 3.

8. According to the Figure 4, why is the error bar size of the experimental results corresponding to ABS larger? And the error bar is missing in Figure 5.

Author Response

(The authors gave the same response as above.)

Round 2

Reviewer 1 Report

The comments have been answered.

Reviewer 2 Report

The manuscript has been well revised.